# Association of Lymphovascular Invasion with Lymph Node Metastases in Prostate Cancer—Lateralization Concept

**DOI:** 10.3390/cancers16050925

**Published:** 2024-02-25

**Authors:** Jakub Karwacki, Adam Gurwin, Arkadiusz Jaworski, Michał Jarocki, Marcel Stodolak, Andrzej Dłubak, Przemysław Szuba, Artur Lemiński, Krystian Kaczmarek, Agnieszka Hałoń, Tomasz Szydełko, Bartosz Małkiewicz

**Affiliations:** 1University Center of Excellence in Urology, Department of Minimally Invasive and Robotic Urology, Wroclaw Medical University, Borowska 213, 50-556 Wroclaw, Poland; adam.gurwin@student.umw.edu.pl (A.G.); arkadiusz.jaworski.lek@gmail.com (A.J.); marcelstodolak@gmail.com (M.S.); andrzej.dlubak@student.umw.edu.pl (A.D.); 2Faculty of Economics in Opole, The WSB University in Wroclaw, Fabryczna 29-31, 53-609 Wroclaw, Poland; przemyslaw.szuba@opole.merito.pl; 3Department of Urology and Urological Oncology, Pomeranian Medical University, Powstańców Wielkopolskich 72, 70-111 Szczecin, Poland; artur.leminski@pum.edu.pl (A.L.); k.kaczmarek.md@gmail.com (K.K.); 4Department of Biochemical Sciences, Pomeranian Medical University, Władysława Broniewskiego 24, 71-460 Szczecin, Poland; 5Department of Clinical and Experimental Pathology, Wroclaw Medical University, Borowska 213, 50-556 Wroclaw, Poland; agnieszka.halon@umw.edu.pl; 6University Center of Excellence in Urology, Wroclaw Medical University, Borowska 213, 50-556 Wroclaw, Poland; tomasz.szydelko@umw.edu.pl

**Keywords:** prostate cancer, radical prostatectomy, lymphovascular invasion, histopathological examination, lymph node invasion, nodal involvement lateralization, pelvic lymph node dissection

## Abstract

**Simple Summary:**

Prostate cancer (PCa) patients often face uncertainties in treatment decisions, particularly regarding lymphadenectomy. This study, involving 96 PCa patients, explores the significance of lymphovascular invasion (LVI) laterality in influencing lymph node invasion (LNI) patterns. Out of these patients, 63.5% exhibited LVI exclusively on the left, 25.0% on the right, and 11.5% on both sides. Significant correlations were observed between LVI laterality and lymph node involvement (*p* < 0.001), especially on the right side. Left-sided LVI correlated with higher cancer stage (*p* = 0.047) and greater odds of bilateral lymph node involvement. This pioneering study emphasizes the need for future prospective, multi-center investigations, ideally incorporating preoperative LVI assessment, to refine PCa treatment decisions.

**Abstract:**

Background. Lymphovascular invasion (LVI) is a vital but often overlooked prognostic factor in prostate cancer. As debates on lymphadenectomy’s overtreatment emerge, understanding LVI laterality gains importance. This study pioneers the investigation into PCa, aiming to uncover patterns that could influence tailored surgical strategies in the future. Methods. Data from 96 patients with both LVI and lymph node invasion (LNI) were retrospectively analyzed. All participants underwent radical prostatectomy (RP) with modified-extended pelvic lymph node dissection (mePLND). All specimens underwent histopathological examination. The assessment of LVI was conducted separately for the right and left lobes of the prostate. Associations within subgroups were assessed using U-Mann–Whitney and Kruskal–Wallis tests, as well as Kendall’s tau-b coefficient, yielding *p*-values and odds ratios (ORs). Results. Out of the 96 patients, 61 (63.5%) exhibited exclusive left-sided lymphovascular invasion (LVI), 24 (25.0%) had exclusive right-sided LVI, and 11 (11.5%) showed bilateral LVI. Regarding nodal involvement, 23 patients (24.0%) had LNI solely on the left, 25 (26.0%) exclusively on the right, and 48 (50.0%) on both sides. A significant correlation was observed between lateralized LVI and lateralized LNI (*p* < 0.001), particularly in patients with right-sided LVI only. LN-positive patients with left-sided LVI tended to have higher pT stages (*p* = 0.047) and increased odds ratios (OR) of bilateral LNI (OR = 2.795; 95% confidence interval [CI]: 1.231–6.348) compared to those with exclusive right-sided LVI (OR = 0.692; 95% CI: 0.525–0.913). Conclusions. Unilateral LVI correlates with ipsilateral LNI in PCa patients with positive LNs, notably in cases of exclusively right-sided LVI. Left-sided LVI associates with higher pT stages and a higher percentage of bilateral LNI cases.

## 1. Introduction

Lymphovascular invasion (LVI), also referred to as microvascular invasion or vessel tumor embolus, is a critical histopathological feature observed in various malignancies [1,2,3,4,5,6,7,8]. In prostate cancer (PCa), the second-most prevalent solid tumor globally, LVI has emerged as a pivotal factor linked to adverse prognostic outcomes [9,10,11,12,13]. In the context of radical prostatectomy (RP), a primary treatment modality for localized PCa, the significance of LVI is underscored by its association with unfavorable clinical outcomes, including lymph node invasion (LNI) [10,12]. Despite LVI being a well-established risk factor for nodal metastasis, there is currently no debate regarding the lateralization of LVI and its potential association with lateralized LNI. To our knowledge, this study represents the first comprehensive analysis of LVI lateralization in PCa, aiming to elucidate its potential correlation with lateralized nodal involvement and explore relevant clinicopathological differences.

Our primary objective is to explore whether the lateralization of LVI, occurring exclusively in the left lobe, right lobe, or both lobes of the prostate, holds significance in terms of LNI. Specifically, we aim to determine if there is an association between the side of LVI and ipsilateral LNI. Additionally, we seek to investigate potential differences in clinicopathological data among patients exhibiting varying patterns of LVI lateralization. By addressing these questions, our study endeavors to contribute novel insights into the intricate relationship between LVI lateralization and nodal metastasis in PCa, paving the way for more informed clinical decision-making in the management of this prevalent malignancy, with implications for the potential application of unilateral lymphadenectomy, particularly in the context of ongoing debates around the feasibility and advantages of this approach [14,15].

## 2. Materials and Methods

### 2.1. Patient Population and Surgical Technique

A cohort of 1016 patients with histologically confirmed PCa undergoing RP at the University Center of Excellence in Urology, Wrocław, Poland, between 2012 and 2022 was analyzed. Exclusion criteria encompassed neoadjuvant therapy, absence of LVI in final histopathology (pL0), negative lymph nodes (pN0), and incomplete clinicopathological data. Following the exclusion criteria, the study ultimately comprised 96 men with positive lymph nodes (LNs) and LVI. Patient selection is depicted in Figure 1. Clinical T staging followed the 2016 TNM classification, with prostate biopsies obtained through transrectal ultrasound (TRUS)-guided systematic, targeted, or combined approaches. Baseline characteristics and clinical parameters were retrospectively collected. Preoperative data included age, preoperative serum PSA levels, biopsy Gleason score (Gleason Grading Groups, GGG), and clinical tumor (cT) stage assessed via digital rectal examination (DRE), bone scintigraphy, and magnetic resonance imaging (MRI). Surgical approaches for RP comprised either open with an ascending technique or laparoscopic with transperitoneal access. Modified-extended pelvic lymph node dissection (mePLND) was conducted, involving the obturator fossa, external, internal, and common iliac vessels, presacral regions, and Marcille’s fossa. A comprehensive description of the lymphadenectomy template was previously published in our other study [16]. Perioperative and histopathological data included pathological T (pT) stage, postoperative Gleason Grading Group (GGG), number of removed LNs, and positive LN count. Excised LNs underwent histopathological examination as separate specimens.

### 2.2. Histopathological Examination

Following the Stanford protocol guidelines, surgical specimens underwent collection and processing. The specimens were fixed in a neutral buffered formalin solution, followed by embedding in paraffin. Utilizing a microtome, tissue samples were sectioned and stained with hematoxylin and eosin (H&E). Experienced uropathologists evaluated slides, adhering to a standardized reporting system. Pathological staging adhered to the American Committee’s guidelines for the Staging System for Prostate Cancer, and Gleason scores were determined in accordance with the International Society of Urological Pathology (ISUP) PCa grading consensus [17,18]. A detailed examination of pathological findings included the assessment of LVI presence, with documentation of laterality—whether LVI was exclusive to the left, right, or both sides of the prostate gland. LVI was defined as the unequivocal presence of tumor cells within endothelial-lined spaces lacking underlying muscular walls or the presence of tumor emboli in small intraprostatic vessels [19,20]. LVI analysis encompassed evaluations in both prostate and seminal vesicles. Within our study cohort, all patients exhibited LVI exclusively in prostate specimens, with no instances of LVI in seminal vesicles. Although the presence of LVI in seminal vesicles was not an exclusion criterion, it is an infrequent occurrence in our center’s experience. In situations of diagnostic ambiguity, podoplanin (D2-40 or PDPN) staining was employed to assist uropathologists in their decision-making process.

### 2.3. Statistical Analysis

Statistical analyses were conducted using PS Imago Pro 9.0 with a Polish license. Continuous variables were presented as means ± standard deviation (SD) or median (range), while categorical variables were expressed as numbers (percentage). The normal distribution of variables was assessed using Shapiro–Wilk tests, revealing a significant deviation from the normal distribution for all analyzed variables [21]. Consequently, nonparametric measures were employed.

To compare mean levels between two groups with categorical variables, the U-Mann–Whitney test was utilized. The Kruskal–Wallis test assessed differences in mean levels among groups with categorical variables, each with at least three levels. Additionally, Kendall’s tau-b coefficient was applied to determine the statistical dependence between two variables. A two-sided testing approach was consistently employed, considering statistically significant differences when the *p*-value was less than 0.05.

Odds ratios (ORs) were calculated to assess the odds for LNI lateralization in subgroups, specifically in patients with LVI exclusively in the left lobe, exclusively in the right lobe, and in both lobes. A confidence interval (CI) of 95% was applied for these calculations.

The utilization of a Marimekko chart was employed as a graphical representation to elucidate associations between LVI and LNI while highlighting their respective ratios. This approach was chosen for its effectiveness in visually conveying the intricate relationships and proportions between these variables, offering a comprehensive and accessible portrayal of the data.

## 3. Results

### 3.1. Patient Population

The mean age of patients at the time of diagnosis was 64.3 years, ranging from 41 to 78 years, and the median prostate-specific antigen (PSA) level was 22.0 ng/mL. Preoperative staging and grading involved the assessment of clinical tumor (cT) stage and Gleason Grading Group (GGG) at biopsy. Clinical examination included both digital rectal examination (DRE) and magnetic resonance imaging (MRI) evaluation. Following histopathological examination of RP specimens, 1 patient (1.0%) had pT2a disease, 5 patients (5.2%) had pT2c disease, 14 patients (14.6%) had pT3a disease, and 76 patients (79.2%) presented with pT3b disease. The mean number of dissected LNs was 21.5 (range: 5–74), while the mean number of positive LNs was 4.2 (range: 1–30). The median percentage of positive LNs (calculated by dividing the number of positive LNs by the total number of resected LNs) was 13.4% (range: 2–100%). LNI was evenly distributed between unilateral and bilateral occurrences, each observed in 48 patients (50%). The tumor location involved both lobes of the prostate in all cases, with varying percentages of tissue occupancy. Table 1 presents the comprehensive clinicopathological data for the entire study population.

### 3.2. Unilateral and Bilateral Lymphovascular Invasion

Unilateral LVI was identified in 85 patients (88.5%), with 61 patients (71.8%) exhibiting LVI exclusively in the left lobe and 24 patients (28.2%) exclusively in the right lobe. Bilateral LVI was observed in 11 patients (11.5%). Regarding pT stage, unilateral LVI was identified in 1 patient (1.2%) with pT2a disease, 5 patients (5.9%) with pT2c disease, 14 patients (16.5%) with pT3a disease, and 65 patients (76.5%) with pT3b cancer. All men with bilateral LVI exhibited pT3b disease in the final histopathology. The mean number of dissected LNs was 21.2 (range: 5–74) in unilateral LVI patients and 23.5 (range: 12–35) in bilateral LVI patients. Additionally, the mean number of positive LNs was 4.1 (range: 1–30) in unilateral LVI patients and 4.4 (range: 1–9) in bilateral LVI patients. The mean percentage of positive LNs was 19.7% (range: 2.4–100%) in unilateral LVI patients and 17.7% (range: 4.5–31.6%) in bilateral LVI patients.

In the group of 96 pL1 pN1 patients included in the study and 38 unincluded pL1 pN0 patients (*n* = 144), 87 (60.4%) had LVI only on the left side, 40 (27.8%) had LVI only on the right, and 16 (11.1%) had LVI on both sides of the prostate.

Table 2 shows the clinicopathological data in the subgroups of 85 unilateral LVI patients and 11 bilateral LVI patients. Figure 2 depicts the patient distribution, classifying them into unilateral left, unilateral right, and bilateral groups based on LVI laterality. Similarly, Figure 3 illustrates the distribution of patients, categorizing them into unilateral left, unilateral right, and bilateral groups based on nodal invasion laterality.

### 3.3. Unilateral Left and Unilateral Right Lymphovascular Invasion

In the cohort of 85 patients with unilateral LVI, the majority, 61 individuals (71.8%), exhibited LVI exclusively in the left lobe, while 24 patients (28.2%) had LVI isolated to the right lobe. Regarding pT stage, unilateral left LVI patients exhibited the following distribution: 2 patients (3.3%) had pT2c disease, 8 patients (13.1%) had pT3a, and 51 (83.6%) had pT3b disease. In unilateral right LVI patients, 1 patient (4.2%) had pT2a disease, 3 patients (12.5%) had pT2c disease, 6 patients (25.0%) had pT3a, and 14 (58.3%) had pT3b. The mean numbers of positive LNs were 4.3 (range: 1–23) and 3.8 (range: 1–30) in the unilateral left and right LVI groups, respectively. A comprehensive presentation of clinicopathological data, including associations with LVI laterality along with corresponding *p*-values, is provided in Table 3.

### 3.4. Odds Ratios and Patient Distribution

Unilateral left LVI patients exhibited an OR of 3.609 (95% CI: 0.925–14.077) for exclusive ipsilateral LNI, an OR of 0.185 (95% CI: 0.092–0.374) for exclusive contralateral LNI, and an OR of 2.795 (95% CI: 1.231–6.348) for bilateral LNI. Meanwhile, unilateral right LVI patients demonstrated an OR of 2.862 (95% CI: 1.531–5.348) for exclusive ipsilateral LNI, an OR of 0.725 (95% CI: 0.579–0.908) for exclusive contralateral LNI, and an OR of 0.692 (95% CI: 0.525–0.913) for bilateral LNI. The detailed ORs and their 95% CIs are outlined in Table 4, while Figure 4 visually represents the proportional relationships in our study cohort through a Marimekko chart.

## 4. Discussion

This study is the first to evaluate the laterality of LVI in PCa and its correlation with the lateralization of nodal involvement. While the impact of LVI on lateralized LNI has been explored in various malignancies such as thyroid, oropharyngeal, and rectal cancers, the specific investigation of LVI laterality in the setting of PCa has not been previously undertaken [22,23,24,25,26,27,28,29]. Existing studies have primarily focused on the broader question of whether LVI influences lateralized LNI, omitting an in-depth exploration of LVI laterality itself [22,23].

LVI is a significant histopathological finding linked to unfavorable outcomes such as biochemical recurrence (BCR), nodal metastases, and other adverse histopathological consequences [30]. Notably, various studies support LVI as an independent factor associated with a worse prognosis. The meta-analysis by Jiang et al. reported a correlation between LVI and BCR (HR = 1.25; 95% CI: 1.17, 1.34; *p* < 0.001, multivariate analysis) as well as nodal involvement (OR = 18.56; 95% CI: 7.82–44.06) [30]. This aligns with findings from previous meta-analyses and large cohort studies [12,13,31,32]. Additionally, existing research has identified associations between LVI and distant metastases [33,34].

The ongoing debate in the PCa field regarding ipsi- and contralateral LNI, coupled with emerging studies on sentinel LN mapping, particularly heightens the relevance of our investigation [15,16,35,36,37]. Bilateral pelvic lymphadenectomy performed during RP remains the gold standard for nodal staging [38]. Despite its debatable therapeutic utility and an increased risk of procedure-associated morbidity, its position in the current guidelines is well-established [38,39,40,41,42,43]. Nonetheless, a growing perspective suggests that a subset of patients may benefit from unilateral PLND. A recent investigation conducted by Martini et al. identified the absence of high-risk disease features as a potential characteristic that might offer substantial benefits to patients undergoing ipsilateral PLND while omitting contralateral LNs [15]. Future investigations could enhance the process of patient selection for unilateral lymphadenectomy by incorporating additional factors. LVI, a histopathological parameter assessable not only in the final histopathology but also preoperatively through prostate biopsy, holds potential for providing valuable insights into the lateralization of PCa progression and metastasis, particularly when considering the distinct assessment of the left and right lobes separately. With the inclusion of preoperative factors that could facilitate early patient selection in the management process, LVI, along with other parameters, could play a significant role in determining candidates for unilateral lymphadenectomy [44]. Moreover, this study holds significance in the era of continually advancing imaging techniques that could offer greater insights into the lateralization concept in PCa [45,46]. Integrating multiple factors (such as LVI, dominant tumor location, or perineural invasion) assessed individually on the right and left sides of the prostate, along with modern imaging techniques, could pave the way for unilateral lymphadenectomy in selected patients. This approach may help avoid the adverse outcomes associated with bilateral PLND, including longer operative time and a higher burden of perioperative complications [39].

Our study revealed that lateral LVI is associated with lateralization of LNI in LN-positive PCa patients (*p* < 0.001). This correlation is particularly pronounced in unilateral right pL1 patients, as over half of patients (35/61; 56.4%) with LVI exclusively on the left side of the prostate exhibited bilateral LNI. Notably, LN-positive patients exhibiting LVI exclusively on the right side appear to manifest a lower risk disease phenotype compared to those with exclusive left-sided LVI. This observation aligns with the identified correlation with pT stage (*p* = 0.047), emphasizing a predilection for exclusively right-sided LNI. However, cautious interpretation of these findings is warranted due to the limited size of the patient cohort and the proximity of the *p*-value confirming the correlation with pT stage to the significance threshold (*p* = 0.047). Furthermore, the dissonance between the left- and right-LVI groups may also be attributed to the complex anatomical lymphatic drainage. Although studies have explored lymphatic drainage patterns and their association with specific lymph node groups, the complexity of intraprostatic lymphatic vessels remains unclear [47,48,49]. Moreover, the obstruction of intraprostatic lymphatic system, and the surgical manipulation itself, could lead to uneven distribution of unilateral right and left LVI patients [50].

In the unilateral left LVI group, the occurrence of ipsilateral LNI was 2.71 times higher (19/7) than contralateral LNI. Conversely, in the unilateral right LVI group, patients with ipsilateral LNI were 8 times more prevalent (16/2) than those with contralateral LNI. Notably, among patients with bilateral nodal metastases (*n* = 48), the majority (35/48, 72.9%) exhibited unilateral left LVI in the prostate gland, while only seven individuals (7/48, 14.6%) had bilateral LVI. Several factors may contribute to these observations, including the relatively small cohort size, and the lack of information on nerve-sparing approaches during surgery, as well as lymphatic system complexity. Nevertheless, we believe that our study can contribute additional insights to the concept of lateralization in PCa nodal metastasis, potentially enriching the discussion on unilateral lymphadenectomy and identifying the patients who could benefit the most from this approach. Perhaps incorporating the laterality of LVI into the existing parameters of the lateralization concept could provide another argument either in favor of or against unilateral PLND in selected patients.

In discussing the limitations of our study, it is imperative to acknowledge the constraints posed by the relatively small patient cohort. The modest sample size may have impacted the statistical power, influencing the significance of *p*-values. This limitation underscores the necessity for further investigations with larger cohorts to validate and strengthen the observed correlations. Additionally, the retrospective and single-center nature of the data collection poses inherent limitations. A multi-center approach and a prospective study design would enhance the generalizability and robustness of the findings. Furthermore, the inclusion of LVI status from biopsy specimens, in addition to the final histopathology, could provide a more comprehensive understanding of the temporal aspects of LVI development and progression. A notable consideration is the absence of data on dominant tumor location, which could offer valuable insights into the laterality issue. Although the tumors in our patient cohort were predominantly located in both lobes, the lack of information on specific tumor locations within the lobes limits our ability to explore the potential impact of tumor localization on LVI laterality. Finally, the racial disparity in PCa diagnosis and management, particularly evident in the contrasting outcomes between African-American and Caucasian men, underscores a crucial aspect of PCa research. Thus, it is important to note that our study’s patient cohort consisted solely of Caucasian men from the Polish population, limiting the generalizability of our findings.

## 5. Conclusions

In this retrospective analysis, we observed a correlation between unilateral LVI and ipsilateral LNI in patients with positive LNs in PCa, particularly pronounced in cases where LVI exclusively occurred in the right prostate lobe. Notably, individuals with LVI restricted to the left side tended to exhibit higher pT stages in our study cohort. To the best of our knowledge, this study represents the first investigation on the laterality of LVI in PCa. However, cautious interpretation is warranted given the study’s limited sample size. Future inquiries should ideally adopt a prospective, multi-center design, encompassing more extensive data on primary tumor location. Moreover, integrating preoperative LVI assessment at biopsy, alongside the standard postoperative evaluation in final histopathology, has the potential to enhance our overall comprehension of PCa progression. Additionally, it could provide valuable insights into preoperative decision-making alterations.

## Figures and Tables

**Figure 1 cancers-16-00925-f001:**
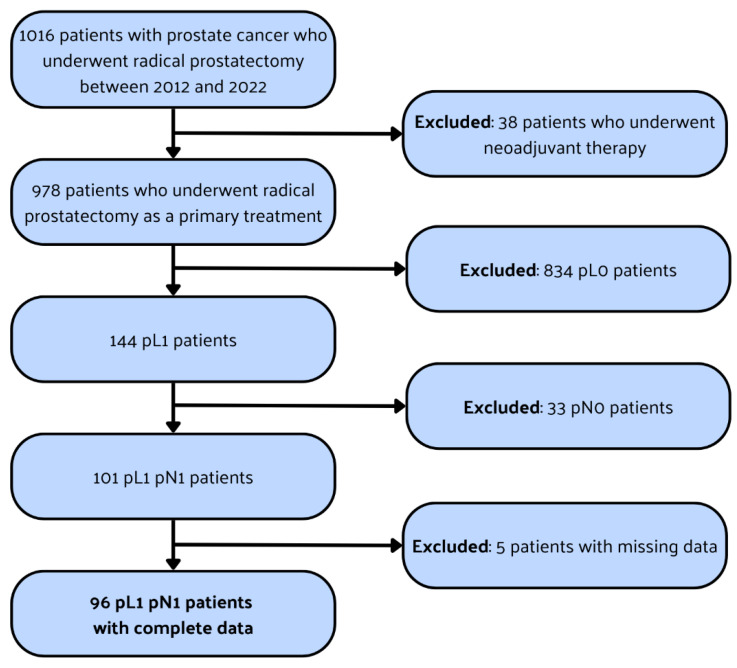
Flowchart illustrating the process of patient selection. pL0: negative lymphovascular invasion (LVI) in histopathological examination; pL1: positive LVI in histopathological examination; pN0: negative lymph node invasion (LNI) in histopathological examination; pN1: positive LNI in histopathological examination.

**Figure 2 cancers-16-00925-f002:**
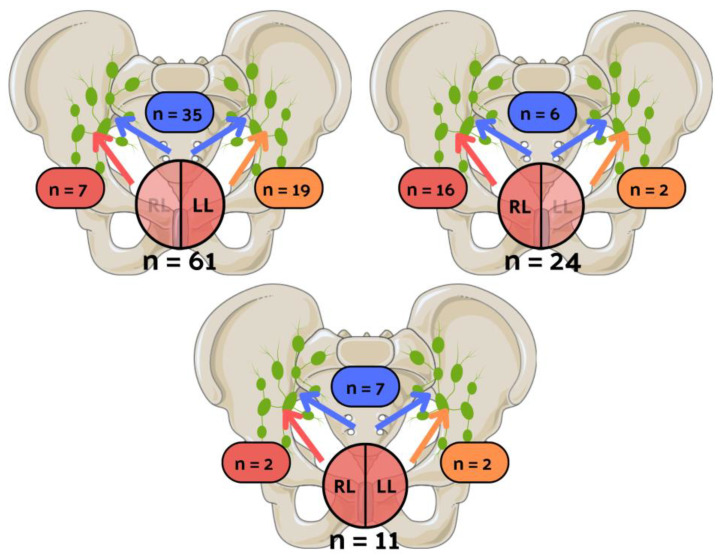
The visual illustration of patients’ distribution, categorizing them into unilateral left, unilateral right, and bilateral groups based on lymphovascular invasion (LVI) laterality. Yellow color represents the number of patients with positive lymph nodes only on the left side, red color represents patients with nodal involvement exclusively on the right side, and blue color represents patients with bilateral nodal invasion. n: number of patients; RL: LVI in the right lobe; LL: LVI in the left lobe.

**Figure 3 cancers-16-00925-f003:**
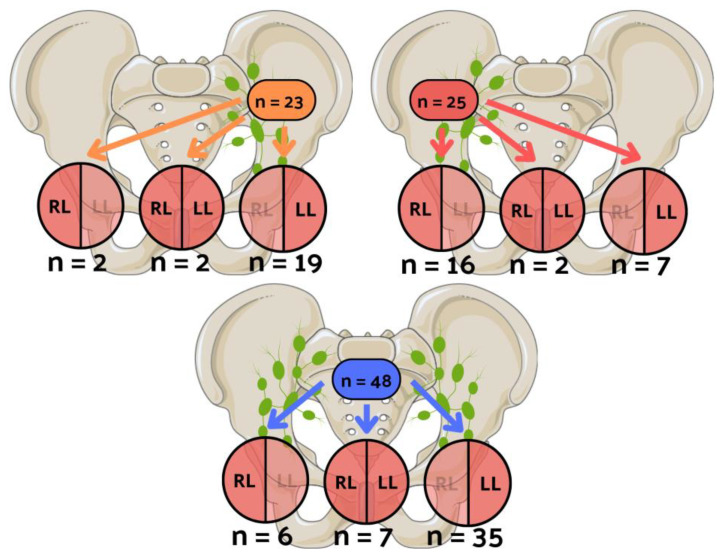
The visual illustration of patients’ distribution, categorizing them into unilateral left, unilateral right, and bilateral groups based on nodal invasion laterality. Yellow color represents the number of patients with positive lymph nodes only on the left side, red color represents patients with nodal involvement exclusively on the right side, and blue color represents patients with bilateral nodal invasion. n: number of patients; RL: lymphovascular invasion (LVI) in the right lobe; LL: LVI in the left lobe.

**Figure 4 cancers-16-00925-f004:**
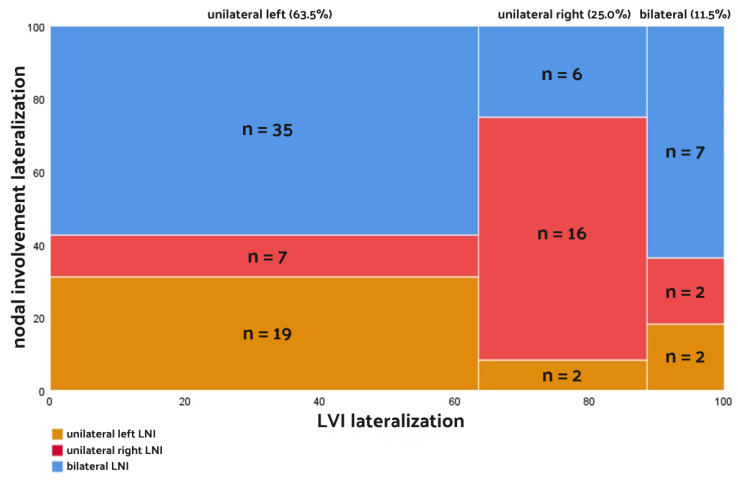
The Marimekko chart depicting the proportional relationship between lymphovascular invasion and lymph node invasion. LVI: lateralization; LNI: lymph node invasion; n: number of patients.

**Table 1 cancers-16-00925-t001:** Characteristics of patient population.

Clinicopathological Data	All Patients (*n* = 96)
Age	64.3 ± 6.8; 64.5 (41–78)
Preoperative PSA	27.9 ± 25.8; 22.0 (2.3–174)
cT stage	
cT1	3 (3.1%)
cT2	52 (54.2%)
cT3	37 (38.5%)
cT4	4 (4.2%)
Biopsy GGG	
1	17 (17.7%)
2	22 (22.9%)
3	22 (22.9%)
4	17 (17.7%)
5	18 (18.8%)
pT stage	
pT2a	1 (1.0%)
pT2c	5 (5.2%)
pT3a	14 (14.6%)
pT3b	76 (79.2%)
Pathological GGG	
1	0 (0.0%)
2	13 (13.5%)
3	30 (31.3%)
4	12 (12.5%)
5	41 (42.7%)
Number of removed LNs	21.5 ± 10.5; 20.0 (5–74)
Number of positive LNs	4.2 ± 4.7; 3 (1–30)
% of positive LNs	19.5 ± 17.2%; 13.4% (2–100%)
LVI laterality	
Unilateral	85 (88.5%)
Bilateral	11 (11.5%)
LNI laterality	
Unilateral	48 (50.0%)
Bilateral	48 (50.0%)

All continuous data are presented as mean ± SD and median (range). All interval data are presented as number and percent. PSA: prostate-specific antigen; cT: clinical tumor stage; GGG: Gleason Grading Group; pT: pathological tumor stage; LNs: lymph nodes; LVI: lymphovascular invasion; LNI: lymph node invasion.

**Table 2 cancers-16-00925-t002:** Comparison of clinicopathological data in patients with unilateral and bilateral lymphovascular invasion.

Clinicopathological Data	Patients with Unilateral LVI (*n* = 85)	Patients with Bilateral LVI (*n* = 11)
Age	64.6 ± 6.5; 64.6 (42–78)	61.3 ± 8.6; 64.2 (41–71)
Preoperative PSA	28.1 ± 24.8; 22.9 (2.3–174.0)	26.3 ± 33.7; 14.3 (7.9–124.9)
Clinical T stage		
cT1	3 (3.5%)	0 (0.0%)
cT2	45 (52.9%)	7 (63.6%)
cT3	34 (40.0%)	3 (27.3%)
cT4	3 (3.5%)	1 (9.1%)
Biopsy GGG		
1	17 (20.0%)	0 (0.0%)
2	17 (20.0%)	5 (45.5%)
3	18 (21.2%)	4 (36.4%)
4	16 (18.8%)	1 (9.1%)
5	17 (20.0%)	1 (9.1%)
Laterality		
Left	61 (71.8%)	-
Right	24 (28.2%)	-
Pathological T stage		
pT2a	1 (1.2%)	0 (0.0%)
pT2c	5 (5.9%)	0 (0.0%)
pT3a	14 (16.5%)	0 (0.0%)
pT3b	65 (76.5%)	11 (100.0%)
Pathological GGG		
1	0 (0.0%)	0 (0.0%)
2	12 (27.1%)	1 (9.1%)
3	23 (27.1%)	7 (63.6%)
4	11 (12.9%)	1 (9.1%)
5	39 (45.9%)	2 (18.2%)
Number of removed LNs	21.2 ± 10.9; 20.0 (5–74)	23.5 ± 7.2; 24.0 (12–35)
Number of positive LNs	4.1 ± 5.0; 3.0 (1–30)	4.4 ± 2.7; 5.0 (1–9)
% of positive LNs	19.7 ± 18.0%; 13.3% (2.4–100%)	17.7 ± 9.2%; 20.0% (4.5–31.6%)

All continuous data are presented as mean ± SD and median (range). All interval data are presented as number and percent. LVI: lymphovascular invasion; PSA: prostate-specific antigen; cT: clinical tumor stage; GGG: Gleason Grading Group; pT: pathological tumor stage; LNs: lymph nodes.

**Table 3 cancers-16-00925-t003:** Comparison of clinicopathological data in patients with unilateral left and unilateral left right lymphovascular invasion.

Clinicopathological Data	Patients with Unilateral Left LVI (*n* = 61)	Patients with Unilateral Right LVI (*n* = 24)	*p*-Value
Age	64.6 ± 6.4; 64.9 (42.2–76.8)	64.8 ± 6.8; 63.8 (54.2–78.0)	0.792
Preoperative PSA	30.2 ± 27.4; 24.2 (2.3–174.0)	22.9 ± 15.9; 19.7 (4.4–76.0)	0.287
Clinical T stage			0.463
cT1	3 (4.9%)	0 (0.0%)	
cT2	31 (50.8%)	14 (58.3%)	
cT3	24 (39.3%)	10 (41.7%)	
cT4	3 (4.9%)	0 (0.0%)	
Biopsy GGG			0.143
1	9 (14.8%)	8 (33.3%)	
2	14 (23.0%)	3 (12.5%)	
3	13 (21.3%)	5 (20.8%)	
4	10 (16.4%)	6 (25.0%)	
5	15 (24.6%)	2 (8.3%)	
Pathological T stage			**0.047**
pT2a	0 (0.0%)	1 (4.2%)	
pT2c	2 (3.3%)	3 (12.5%)	
pT3a	8 (13.1%)	6 (25.0%)	
pT3b	51 (83.6%)	14 (58.3%)	
Pathological GGG			0.464
1	0 (0.0%)	0 (0.0%)	
2	8 (13.1%)	4 (16.7%)	
3	14 (23.0%)	9 (37.5%)	
4	9 (14.8%)	2 (8.3%)	
5	30 (49.2%)	9 (37.5%)	
Number of removed LNs	21.5 ± 10.2; 20.0 (5–67)	20.5 ± 12.6; 18.5 (9–74)	0.379
Number of positive LNs	4.3 ± 4.5; 3.0 (1–23)	3.8 ± 6.2; 2.0 (1–30)	0.069
% of positive LNs	20.8% ± 18.5%; 16.7% (2.4–100%)	16.9% ± 16.7%; 10% (3.6–61.1%)	0.135
LNI laterality			**<0.001**
Unilateral left	19 (31.1%)	2 (8.3%)	
Unilateral right	7 (11.5%)	16 (66.7%)	
Bilateral	35 (56.4%)	6 (25.0%)	

All continuous data are presented as mean ± SD and median (range). All interval data are presented as number and percent. LVI: lymphovascular invasion; PSA: prostate-specific antigen; cT: clinical tumor stage; GGG: Gleason Grading Group; pT: pathological tumor stage; LNs: lymph nodes.

**Table 4 cancers-16-00925-t004:** Odds ratios and 95% confidence intervals illustrating the associations between lymphovascular invasion laterality and lymph node invasion laterality in prostate cancer patients.

LNI Laterality	Patients with Unilateral Left LVI (*n* = 61)	OR (95% CI)	Patients with Unilateral Right LVI (*n* = 24)	OR (95% CI)
Unilateral left	19 (31.1%)	3.609 (0.925–14.077)	2 (8.3%)	0.725 (0.579–0.908)
Unilateral right	7 (11.5%)	0.185 (0.092–0.374)	16 (66.7%)	2.862 (1.531–5.348)
Bilateral	35 (57.4%)	2.795 (1.231–6.348)	6 (25.0%)	0.692 (0.525–0.913)

LNI: lymph node invasion; LVI: lymphovascular invasion; n: number of patients; OR: odds ratio; CI: confidence interval.

## Data Availability

The data supporting the findings of this study are available upon request. Please contact the corresponding author for access to the dataset.

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
