# Peer review of "Association of Lymphovascular Invasion with Lymph Node Metastases in Prostate Cancer—Lateralization Concept"

_cancers, 2024, doi:10.3390/cancers16050925_

Round 1
Reviewer 1 Report
Comments and Suggestions for Authors
The paper entitled “Association of Lymphovascular Invasion with Lymph Node 2 Metastases in Prostate Cancer – Lateralization Concept” provides some interesting retrospective clinical observations and analysis from 96 patients with positive lymphovascular invasion (LVI) and lymph node invasion (LNI) from 1016 cases with radical prostatectomy (RP). LVI has been a strong predictor for lymph node invasion (LNI), disease progression and even overall survival of several cancers including prostate cancer. There is also a unilateralized correlation between these two pathological features, LVI and LNI. The ipsilateral concurrence with a potential uncertainty leading to contralateral LNI has excited the debates on the clinical practice of lymph node dissection on both sides during radical prostatectomy. This warrants further studies and more data to clarify the risk the clinical practices may cause for lymphadenectomy decision making. In this study a significant correlation was observed between lateralized LVI and lateralized LNI, particularly in patients with right-sided LVI only. Patients with left-sided LVI tended to have higher pT stages and increased bilateral LNI compared to those with exclusive right-sided. All these data indicate discrepancy of pathological laterization that deserves more investigation on the lateralized heterogeneity and underlying causes. Some minor issues may need further clarification.
1. Line 34-35, the analysis method was missed and may be specified here in abstract.
2. For the Harrison et al. (Cancer. 1992, 69(3):750-4.) has shown the ipsilateral association of palpable tumor side and pelvic lymph node invasion in localized prostate cancer as early as 1992.
3. Have the authors looked into the LV density and the LNI correlation? Is there still lateralized tendency?
4. Would the higher portion of bilateral LNI in left-side LNI patients be related to higher grade tumors?
Author Response
The response to the Reviewer is included in the attached PDF file.

Reviewer 2 Report
Comments and Suggestions for Authors
The manuscript titled "Association of Lymphovascular Invasion with Lymph node 2 Metastases in Prostate Cancer – Lateralization Concept" explores the correlation between lymphovascular invasion (LVI) and lymph node metastasis in prostate cancer patients. It meticulously details patient selection, study methodology, and results, along with a comprehensive discussion. The study contributes new insight into the prognostic factors affecting prostate cancer progression and the potential implications for surgical planning and patient management.
Major points:
1) While the paper carefully and on multiple occasions acknowledges that the patient sample is limited to LVI and LN+ patients, the exclusion of the 33 patients with pN0 from the study seems to somehow hinder the overall purpose of the study. By removing this population from the denominator, many of the OR and CI - although still accurate in the strict context of this study - lose their validity for drawing clinical conclusions. On many occasions, one wonders why pN0 is excluded and how it would affect the overall data on laterality.
2) Along the same theme as the previous point, the lingering initial question that this study gets very close to but fails to address is if the laterality of LVI affects the rate of LN metastasis.
3) The study acknowledges its limitations, one of the more important of which is the laterality of the dominant nodule. This brings up the important question of correlation and causality. In this study, we cannot conclude whether the observed correlation is an independent finding or whether both LVI and LN metastasis are in fact functions of tumor burden, stage, and laterality.
4) Please mention if the diagnosis of LVI is limited to instances inside prostate tissue, or cases with LVI in seminal vesicles are also counted, in which case a pT3b staging may be implied by LVI alone, explaining the very high percentage (79.2%) of pT3b in this cohort.
Minor points:
line 39: clarify the sentence "Patients with left-sided LVI tended to have higher pT stages (p = 0.047)" by mentioning it is in patients with positive lymph node metastasis.
Line 191: Please change the median to mean to conform the data presentation with the rest of the paragraph.
Line 257: Please provide a reference for this statement.
Line 296: For this sentence, "Notably, patients exhibiting LVI exclusively 295 on the right side appear to manifest a lower risk disease phenotype compared to those 296 with exclusive left-sided LVI." Please clarify: in the patient with positive LVI and LN met.
Line 308: "In the unilateral left LVI group, the occurrence of ipsilateral LNI was 2.71 times 308 higher (19/7) than contralateral LNI." This cannot be concluded from the present data without having the full denominator, which includes pN0.
Author Response

(The authors gave the same response as above.)

Reviewer 3 Report
Comments and Suggestions for Authors
We acknowledge the importance of the study's focus on prostate cancer.
However, please highlight a crucial aspect that needs further attention – the racial disparity in prostate cancer, particularly the significant differences observed between African-American and Caucasian men .The statistics in USA indicate that African-American men are 1.76 times more likely to be diagnosed and 2.14 times more likely to die from prostate cancer compared to Caucasian men.
Could the authors address this racial disparity within the context of their study.
Author Response

(The authors gave the same response as above.)

Reviewer 4 Report
Comments and Suggestions for Authors
The authors analyzed relation with LVI laterality and side specific LNI involvements in prostate cancer. This reviewer requested some revisions before accepting the article.
l• In order to select unilateral or bilateral lymph node dissection during radical prostatectomy, biopsy information together with radiographical features should be needed. As the authors pointed out, biopsy LVI information should be presented.
l• Genitourinary pathologists should be included in the authors list.
l• The authors should present cases with positive LVI but negative LNI (pN0). We should like to know how LVI positivity correlated with LNI. Moreover, the authors should also present cases with negative LVI and positive LNI. Presenting all these data, we can consider and discuss importance of relation with LVI and LNI.
l• What does “% of positive LNs” mean? Does it mean % volume of cancer involvement among all volume of lymph nodes? Or does it mean positive number of LNs / all resected LNs? Please clarify the issue.
l• Lymph node dissection during radical prostatectomy provide us information about progressive status of prostate cancer patients. Although LVI laterality can predict side specific LNI involvements in most cases, this does not mean that we can omit contralateral lymph node dissection. There are some patients with bilateral LNI or only contralateral LNI even if those patients present only unilateral LVI. How do you explain the issue?
Author Response

(The authors gave the same response as above.)

Round 2
Reviewer 2 Report
Comments and Suggestions for Authors
Thank you for addressing the concerns raised in the review.